# Update of safety profile of bile acid sequestrants: A real-world pharmacovigilance study of the FDA adverse event reporting system

Wangqi Chen[1,2], Yuxia Xie[1,2], Qinghua Li[1,2], Zhenghui Zhu[1,2], Xinyan Li[1,2], Hong Zhu🄳[1,2*]

1 Department of Gastroenterology, The First Affiliated Hospital of Nanjing Medical University, Nanjing, Jiangsu Province, China, 2 Department of Gastroenterology, The First School of Clinical Medicine of Nanjing Medical University, Nanjing, Jiangsu Province, China

* zhuhong1059@126.com

## Abstract

### Background

Bile acid sequestrants (BASs), including cholestyramine, colestipol, and colesevelam, are widely used in endocrine and gastrointestinal disorders. However, their long-term safety remains under-characterized. This study leveraged real-world pharmacovigilance data to evaluate underreported and subclass-specific adverse events (AEs) associated with BASs.

### Methods

We analyzed 5,286 AE reports related to BASs from the FDA Adverse Event Reporting System (2004–2024) using four disproportionality methods: Reporting Odds Ratio (ROR), Proportional Reporting Ratio (PRR), Bayesian Confidence Propagation Neural Network (BCPNN), and Multi-item Gamma Poisson Shrinker (MGPS). AE signals were assessed at both the System Organ Class (SOC) and Preferred Term (PT) levels. Time-to-onset (TTO) analysis was also performed.

### Results

All three BASs showed prominent gastrointestinal AEs. Cholestyramine was notably associated with oropharyngeal irritation (e.g., throat irritation, ROR = 21.89; oropharyngeal discomfort, ROR = 36.53), while colestipol presented mechanical risks such as dysphagia (ROR = 21.51) and choking (ROR = 67.44). Colesevelam exhibited musculoskeletal toxicity, including myalgia (ROR = 4.74) and muscle spasms (ROR = 3.43). Consensus signals across all methods further revealed novel AEs such as dysgeusia, dental abnormalities, gastroesophageal reflux disease, and fecaloma. TTO analysis showed that most AEs occurred within the first month of therapy, with 15–16% persisting beyond 6 months.

**Data availability statement:** All relevant data are within the manuscript and its Supporting Information files.

**Funding:** The author(s) received no specific funding for this work.

**Competing interests:** The authors have declared that no competing interests exist.

## Conclusion

This large-scale FAERS study updates the safety profiles of BASs, highlighting distinct risk patterns and delayed complications. The findings support personalized monitoring strategies that consider both drug-specific characteristics and temporal AE patterns.

## Introduction

Bile acid sequestrants (BASs) are a class of anionic exchange resins that bind bile acids in the intestine and disrupt their enterohepatic circulation, with primary representatives including cholestyramine, colestipol, and colesevelam [1]. These agents are widely used in the treatment of various endocrine and gastrointestinal disorders. Their core mechanism involves promoting hepatic cholesterol catabolism into bile acids and reducing serum low-density lipoprotein cholesterol (LDL-C) concentrations, rendering them suitable for patients intolerant to statins or requiring intensified lipid-lowering therapy [2]. Additionally, colesevelam has been applied in diabetes management due to its glucose-lowering effects via pathways such as stimulating glucagon-like peptide-1 (GLP-1) secretion [3]. In hepatobiliary diseases, BASs mitigate toxic bile acid accumulation, serving as therapeutic options for bile acid diarrhea, cholestatic pruritus, primary biliary cholangitis (PBC), and primary sclerosing cholangitis (PSC) [4–7]. Recent studies further suggest their potential roles in alleviating inflammatory bowel disease (IBD), irritable bowel syndrome (IBS), non-alcoholic fatty liver (NAFL), and microscopic colitis [8–11].

Despite their favorable short-term safety profiles in randomized controlled trials (RCTs), with adverse effects predominantly gastrointestinal, long-term risks remain contentious [12]. Post-marketing surveillance and case reports indicate underappreciated risks, including hypoglycemia, electrolyte imbalances, malabsorption of fat-soluble vitamins, drug interactions, and rare but severe complications such as intestinal obstruction [13–17]. In patients with chronic liver disease or IBD, gastrointestinal adverse effects may overlap with baseline symptoms, exacerbating reduced adherence and complicating therapeutic decisions [8]. However, existing research prioritizes cardiovascular risk-benefit analyses (e.g., LDL-C reduction versus drug interactions) while underexploring hepatobiliary-specific adverse events (AEs) such as hepatotoxicity or gut microbiota dysbiosis [18]. Furthermore, clinical trials are constrained by stringent inclusion/exclusion criteria, limited sample sizes, and short observational periods, hindering comprehensive evaluation of rare or delayed real-world events. As the indications for BASs expand and combination therapies become more common, their safety profiles urgently require updating through real-world data and systematic investigations to optimize risk-benefit balancing in clinical practice.

Post-marketing pharmacovigilance predominantly relies on spontaneous reporting systems to monitor real-world AE patterns and ensure ongoing drug safety evaluation. Among these systems, the FDA Adverse Event Reporting System (FAERS), a large global database for drug-related AEs, aggregates reports from healthcare

professionals, pharmaceutical manufacturers, and patients, offering advantages in real-time updates and broad-spectrum coverage [19]. Through data mining techniques, FAERS enables the effective identification of signals that may be overlooked by traditional research methods, particularly in detecting low-incidence events and temporal drug-event associations. To address these gaps, this study was designed to comprehensively re-evaluate the safety profiles of BASs using data from the FAERS. We first sought to update the spectrum and reporting frequency of AEs associated with the three currently approved BASs—cholestyramine, colestipol, and colesevelam—based on real-world pharmacovigilance data. In addition to known AEs, the study aimed to uncover potentially novel or underreported safety signals that have not been well characterized in pre-marketing trials or post-marketing surveillance. Furthermore, by comparing the AE profiles of individual agents across different system organ classes and preferred terms, we aimed to delineate subclass-specific risk patterns and temporal trends. Through this multi-layered analysis, we hope to provide actionable insights for clinical decision-making and long-term pharmacovigilance strategies.

## Materials and methods

### Data source

FAERS is a global, publicly accessible, and spontaneous reporting database that provides comprehensive real-world data on drug adverse reactions (ADRs). The datasets in FAERS comprise seven distinct subfiles: demographic records (DEMO), drug information (DRUG), adverse event reports (REAC), patient outcomes (OUTC), report sources (RPSR), therapy start/end dates (THER), and indications for use (INDI) [20]. To retrospectively investigate AEs associated with BASs (including cholestyramine, colestipol, and colesevelam) and comprehensively evaluate their safety profiles, we extracted corresponding quarterly datasets from FAERS. All data were processed in the subsequent step using R software (version 4.4.3).

### Data extraction and processing

To address data redundancy inherent in the FAERS database, deduplication was performed following FDA-recommended procedures. For the DEMO dataset, CASEID served as the primary deduplication criterion. When duplicate CASEIDs were identified, we retained records with the most recent FDA_DT. If both CASEID and FDA_DT were identical, entries were further prioritized by PRIMARYID. Initial extraction yielded 22,249,476 raw entries, which were reduced to 18,627,667 records after deduplication. AE reports potentially related to the three BASs were then extracted by screening the DRUG dataset using both generic and brand names of FDA-approved formulations, with the role of drugs designated as the primary suspect (PS). The inclusion of cholestyramine, colestipol, and colesevelam was based on their FDA approval status and current clinical indications. To ensure accurate drug-event attribution, only cases in which these BASs were labeled as the PS were retained. Both generic names and all associated brand names were used for comprehensive identification across FAERS records. Reports where BASs were coded as concomitant or secondary suspect drugs were excluded to reduce signal dilution and confounding. The identified AEs were rigorously standardized and classified according to Preferred Terms (PTs) and System Organ Classes (SOCs) from the Medical Dictionary for Regulatory Activities (MedDRA version 26.1) [21]. To evaluate temporal associations, the time-to-onset (TTO) of BAS-related AEs was quantitatively determined as the interval between drug use initiation (START_DT in THER) and AE occurrence (EVENT_DT in DEMO) for analysis [22]. Comprehensive data collection of patient demographics and relevant AE characteristics was performed for descriptive statistical analysis, including systematic aggregation of variables such as patient sex, age, weight, reporter category, reporting dates, outcomes, and country-level reporting distribution.

### Statistical analysis

Disproportionality analysis, a cornerstone methodology in pharmacovigilance research, enables the quantification of association strength between specific medications and particular AEs through multiple statistical models, thereby identifying

drug safety risks. The disproportionality analysis was conducted using a fourfold table framework, where observed AE-reporting frequencies for bile acid sequestrants versus all other drugs were structured according to the schema detailed in S1 Table. In this study, four disproportionality analysis methods were implemented to detect potential signals between BASs and AEs: ROR, PRR, BCPNN, and MGPS [23]. These four methods were selected due to their complementary characteristics. ROR and PRR are frequentist methods commonly used for signal generation due to their simplicity and interpretability. BCPNN and MGPS, in contrast, adopt Bayesian frameworks that offer better performance in low-count scenarios by incorporating prior distributions and smoothing estimations. By integrating both types, we aimed to increase signal detection robustness while minimizing false positives. Consensus signals identified across all four algorithms were considered more reliable. Signal thresholds were defined according to standard criteria: ROR and PRR were considered positive when the lower bound of the 95% confidence interval exceeded 1 and the number of co-reports was ≥ 3; BCPNN signals required IC025 > 0; and MGPS signals required EBGM05 > 2. Detailed formulas and signal detection thresholds for these algorithms are provided in S2 Table. The combined application of these methods enhances analytical robustness by compensating for individual limitations, thereby improving signal detection reliability while minimizing false-positive risks. A schematic overview of the data extraction, processing, and analytical workflow is illustrated in Fig 1.

## Results

### Clinical characteristics of BAS-related AEs in FAERS

After deduplication, 5,286 AE reports associated with BASs were analyzed, including 1,576 cases for cholestyramine, 616 for colestipol, and 3,094 for colesevelam. Gender distribution revealed that the proportion of female patients was significantly higher than that of males across all three agents. Age distribution analysis highlighted a substantial representation

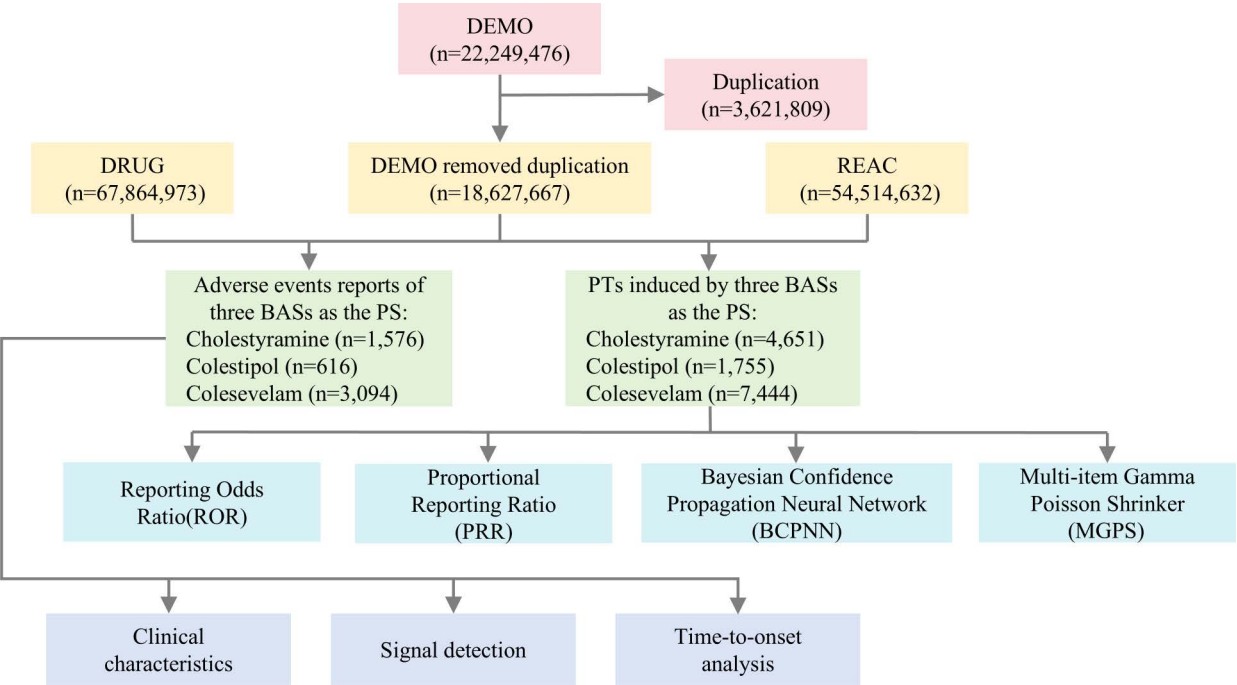

**Fig 1. Schematic diagram of data extraction, processing, and analysis.** DEMO: demographic records, DRUG: drug information, REAC: adverse event reports.

of elderly patients: the 65–85-year-old group constituted the largest proportion for cholestyramine (23.7%), colestipol (31.5%), and colesevelam (14.9%). Notably, colestipol exhibited the highest proportion of patients aged >85 years (4.5%). However, the high rate of missing age data warrants caution in interpreting age-related risk patterns. Regarding reporter categories, consumer-submitted reports accounted for the majority of cholestyramine (69.0%) and colestipol (72.1%) cases, whereas colesevelam had the highest proportion of physician-reported cases (29.5%) among the three agents. Geographically, over 90% of reports originated from the United States. Detailed demographic and reporting characteristics are presented in Table 1. Temporal trends in AE reporting revealed distinct peaks: colesevelam-associated reports peaked in 2015 (n = 1,014), while cholestyramine and colestipol reached their highest reporting frequencies in 2019 (n = 211) and 2018 (n = 107), respectively (Fig 2).

## Detection of signals related to three BASs based on disproportionality analysis

**Signal detection at the SOC level.** The selected AEs were standardized and classified according to the SOCs in MedDRA. The results indicated that safety signals associated with the three BASs spanned 27 common SOC systems. Gastrointestinal Disorders emerged as the most frequently reported category (cholestyramine: 24.1%, colestipol: 21.9%, colesevelam: 19.6%), followed by Injury, Poisoning And Procedural Complications, and General Disorders And Administration Site Conditions. Besides, Musculoskeletal And Connective Tissue Disorders, as well as Metabolism And Nutrition Disorders, both identified in this study, align with common adverse reactions listed in the prescribing information of these agents, supporting the plausibility of our findings. Additionally, the analysis uncovered safety signals not documented in the drug labels, including respiratory, neurological, and dermatological disorders. These observations

**Table 1. Basic characteristics of adverse event reports related to Cholestyramine, Colestipol, and Colesevelam from FAERS.**

| Characteristics | Cholestyramine | Colestipol | Colesevelam |
|---|---|---|---|
| **Number of reports** | 1576 | 616 | 3094 |
| **Gender** | | | |
| Female | 971 (61.6%) | 395 (64.1%) | 1750 (56.6%) |
| Male | 397 (25.2%) | 169 (27.4%) | 742 (24.0%) |
| Unknown | 208 (13.2%) | 52 (8.4%) | 602 (19.5%) |
| **Age** | | | |
| <18 | 10 (0.6%) | 1 (0.2%) | 12 (0.4%) |
| 18-64 | 290 (18.4%) | 115 (18.7%) | 439 (14.2%) |
| 65-85 | 374 (23.7%) | 194 (31.5%) | 462 (14.9%) |
| >85 | 34 (2.2%) | 28 (4.5%) | 40 (1.3%) |
| Unknown | 868 (55.1%) | 278 (45.1%) | 2141 (69.2%) |
| **Reporter** | | | |
| Physician | 200 (12.7%) | 37 (6.0%) | 914 (29.5%) |
| Pharmacist | 147 (9.3%) | 83 (13.5%) | 80 (2.6%) |
| Health-professional | 36 (2.3%) | 11 (1.8%) | 41 (1.3%) |
| Other health-professional | 58 (3.7%) | 27 (4.4%) | 132 (4.3%) |
| Consumer | 1088 (69.0%) | 444 (72.1%) | 1234 (39.9%) |
| Other groups | 0 (0) | 1 (0.2%) | 2 (0.1%) |
| Unknown | 47 (3.0%) | 13 (2.1%) | 691 (22.3%) |
| **Reported Countries** | | | |
| The United States | 1370 (86.9%) | 582 (94.5%) | 2929 (94.7%) |
| Other | 206 (13.1%) | 34 (5.5%) | 165 (5.3%) |

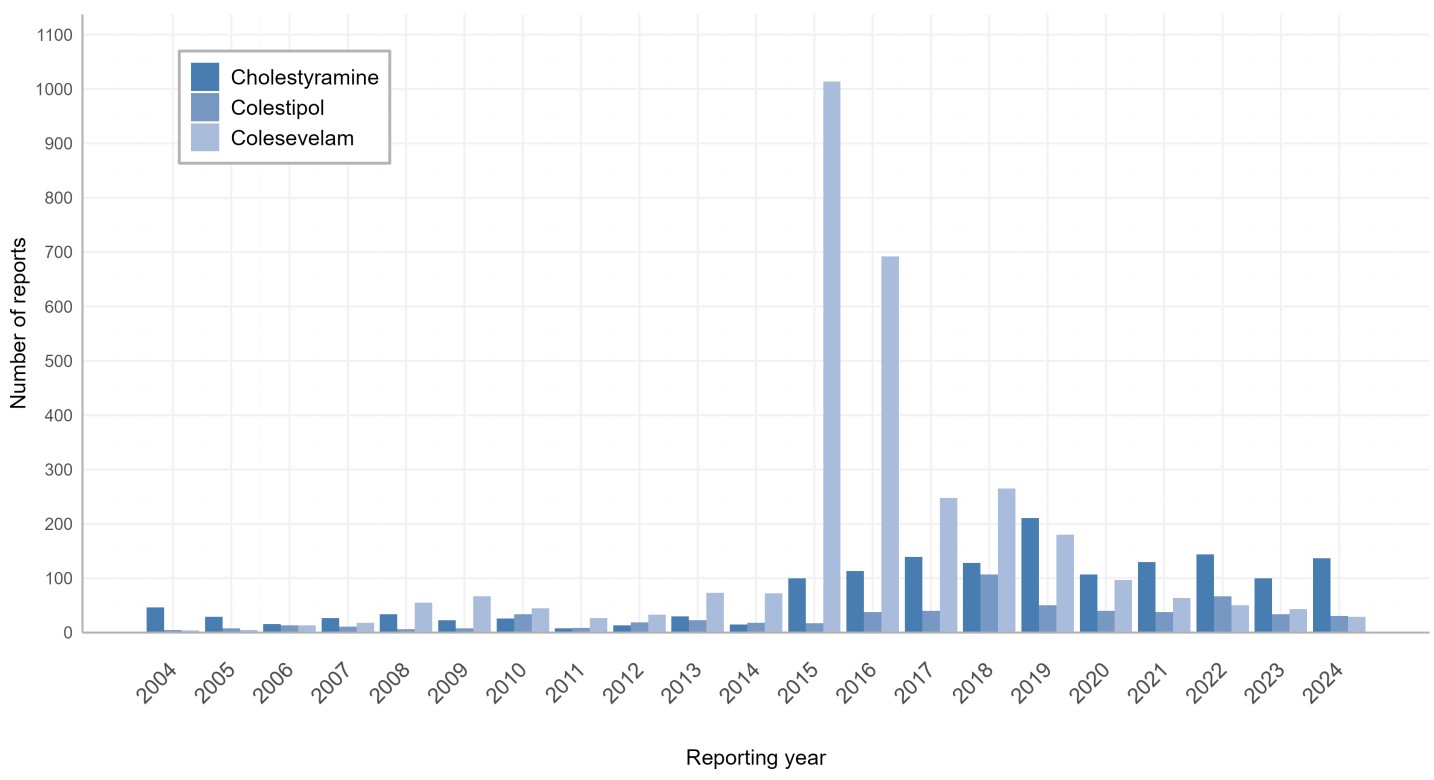

**Fig 2. Temporal trends in adverse event reporting for bile acid sequestrants.**

necessitate further investigation and pharmacovigilance monitoring to comprehensively characterize and update the safety profiles of BASs. The distribution of AEs across SOC categories is illustrated in Fig 3.

**Signal detection at the PT level.** We further investigated the AE signals of the three BASs at the PT level. The top 30 PTs for each BAS, ranked by descending report counts, are shown in Figs 4–6. Gastrointestinal symptoms dominated across all three agents, with strong signals including diarrhea, constipation, abdominal pain, flatulence, and dyspepsia. Cholestyramine exhibited localized irritant risks characterized by high-intensity signals for throat irritation (ROR = 21.89, n = 73), oropharyngeal discomfort (ROR = 36.53, n = 23), and choking sensation (ROR = 67.35, n = 28), along with dysgeusia (ROR = 7.26, n = 43)—none mentioned in its prescribing information. Colestipol demonstrated disproportionately higher mechanical complications, particularly dysphagia (ROR = 21.51, n = 57) and choking (ROR = 67.44, n = 37). Colesevelam showed unique musculoskeletal signals: myalgia (ROR = 4.74, n = 99) and muscle spasms (ROR = 3.43, n = 78) were frequent yet unlisted in labeling, alongside novel signals like gastroesophageal reflux disease (ROR = 4.01, n = 39).

To evaluate the robustness of detected AE signals in the presence of incomplete sex data, we performed sex-stratified disproportionality analyses for the top PTs of each BAS. As shown in S3 Table, both male and female subgroups demonstrated consistent signal patterns. For instance, constipation and dysphagia remained strong signals for colestipol in both males (ROR = 17.19, PRR = 16.74) and females (ROR = 12.94, PRR = 12.72). Similarly, cholestyramine showed persistent signals for throat irritation and retching in both sexes (female ROR = 22.86; male ROR = 81.86). Colesevelam also exhibited reproducible signals for myalgia and muscle spasms across genders. While some variations in ROR magnitudes were observed (likely due to differences in report counts), the core AEs retained statistical significance in both subgroups. These results support the stability of our main findings and mitigate concerns about sex-related reporting bias.

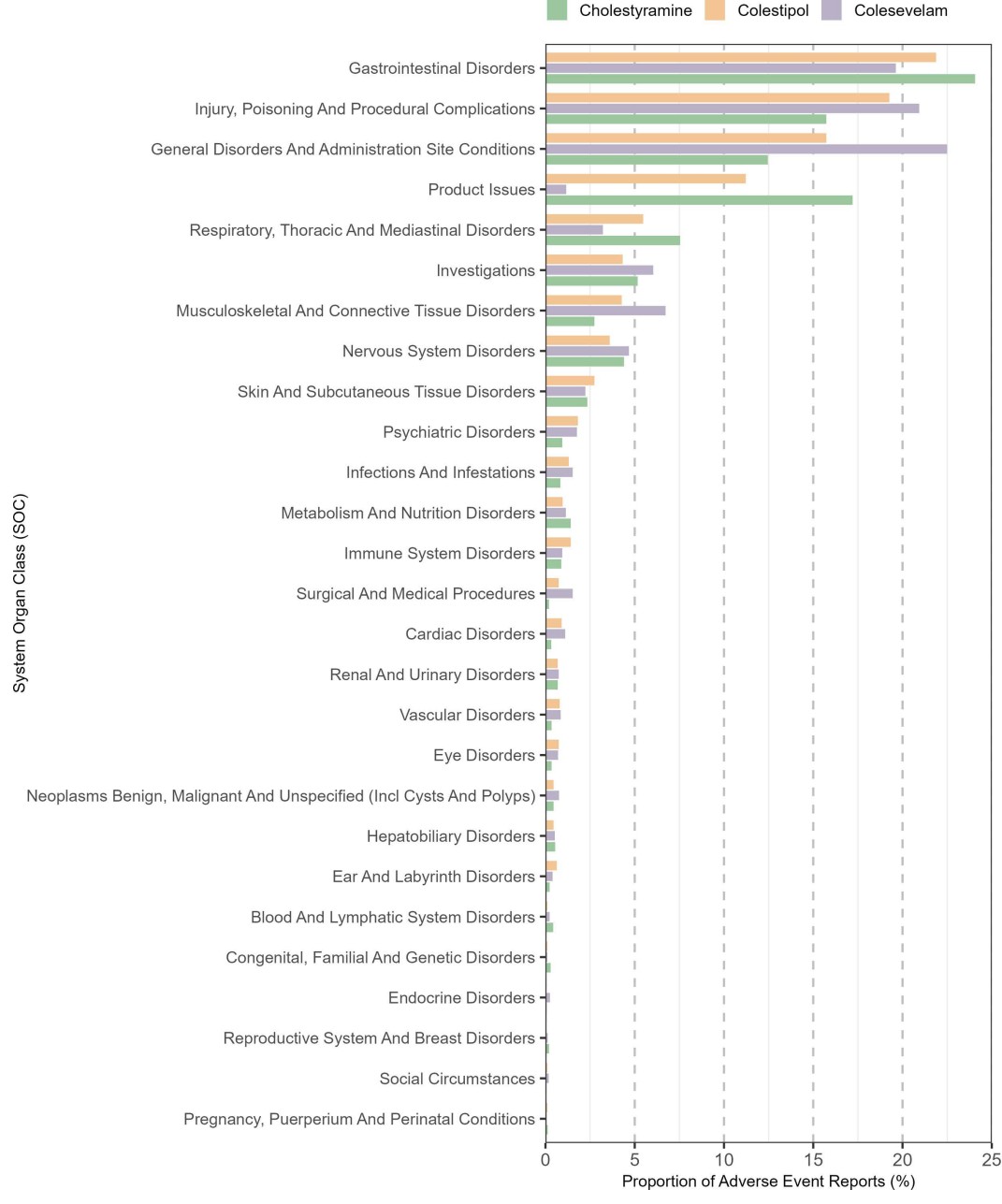

**Fig 3. Distribution of adverse events by MedDRA System Organ Class for three bile acid sequestrants.**

Using four disproportionality analyses (ROR, PRR, BCPNN, MGPS), we identified consensus positive signals through intersection plots (Fig 7). Cholestyramine had 81 PTs (60.4%) concordant across all methods, colestipol 34 PTs (45.3%), and colesevelam 76 PTs (58.5%). Following the exclusion of PTs associated with disease progression or underlying conditions, the number of PTs retained was as follows: cholestyramine (n = 46), colestipol (n = 14), and colesevelam (n = 33). As detailed in S4 Table, novel findings included cholestyramine-associated vitamin K deficiency, intestinal crystal deposits, and dental abnormalities, while colesevelam showed signals for esophageal obstruction and fecaloma.

| SOC | PT(Preffered Term) | a | ROR(95% CI) | |
|-----|------|---|-----|---|
| Gastrointestinal Disorders | Diarrhoea | 198 | 4.17(3.61~4.81) | |
| Gastrointestinal Disorders | Nausea | 104 | 1.73(1.43~2.1) | |
| Gastrointestinal Disorders | Constipation | 94 | 5.91(4.82~7.25) | |
| Respiratory, Thoracic And Mediastinal Disorders | Throat Irritation | 73 | 21.89(17.37~27.59) | |
| Gastrointestinal Disorders | Abdominal Pain Upper | 72 | 4.65(3.68~5.86) | |
| Gastrointestinal Disorders | Abdominal Discomfort | 62 | 4.9(3.82~6.3) | |
| Gastrointestinal Disorders | Retching | 61 | 38(29.5~48.94) | |
| Gastrointestinal Disorders | Vomiting | 57 | 1.6(1.23~2.08) | |
| Respiratory, Thoracic And Mediastinal Disorders | Cough | 46 | 2.17(1.62~2.9) | |
| General Disorders And Administration Site Conditions | Malaise | 46 | 1.33(1~1.78) | |
| Nervous System Disorders | Dysgeusia | 43 | 7.26(5.37~9.8) | |
| Respiratory, Thoracic And Mediastinal Disorders | Choking | 42 | 28.54(21.06~38.69) | |
| Nervous System Disorders | Headache | 42 | 0.86(0.64~1.17) | |
| Gastrointestinal Disorders | Flatulence | 39 | 9.12(6.65~12.5) | |
| Respiratory, Thoracic And Mediastinal Disorders | Oropharyngeal Pain | 35 | 4.93(3.54~6.88) | |
| Gastrointestinal Disorders | Dyspepsia | 32 | 4.31(3.04~6.1) | |
| Gastrointestinal Disorders | Abdominal Distension | 32 | 4.07(2.87~5.76) | |
| Gastrointestinal Disorders | Abdominal Pain | 31 | 1.74(1.22~2.47) | |
| Respiratory, Thoracic And Mediastinal Disorders | Choking Sensation | 28 | 67.35(46.4~97.76) | |
| Investigations | Blood Glucose Increased | 28 | 1.88(1.29~2.72) | |
| Investigations | Weight Decreased | 28 | 1.3(0.89~1.88) | |
| Skin And Subcutaneous Tissue Disorders | Pruritus | 25 | 0.91(0.62~1.35) | |
| Skin And Subcutaneous Tissue Disorders | Rash | 25 | 0.76(0.51~1.13) | |
| Respiratory, Thoracic And Mediastinal Disorders | Oropharyngeal Discomfort | 23 | 36.53(24.24~55.07) | |
| Gastrointestinal Disorders | Dysphagia | 22 | 3.04(2~4.63) | |
| Nervous System Disorders | Dizziness | 22 | 0.57(0.37~0.86) | |
| Respiratory, Thoracic And Mediastinal Disorders | Dyspnoea | 22 | 0.5(0.33~0.76) | |
| General Disorders And Administration Site Conditions | Fatigue | 22 | 0.37(0.24~0.56) | |
| Musculoskeletal And Connective Tissue Disorders | Arthralgia | 21 | 0.66(0.43~1.01) | |
| Immune System Disorders | Hypersensitivity | 19 | 1.33(0.85~2.08) | |

**Fig 4. Adverse event signal distribution of cholestyramine at Preferred Term level.** Top 30 PTs ranked by report frequency. SOC, System Organ Class; a, number of reports containing both the target drug and the target adverse drug reaction; ROR, Reporting Odds Ratio.

## Time-to-onset analysis of BAS-related AEs

We conducted a time-to-onset analysis of AEs associated with BASs, restricted to reports with clearly documented AE timelines. A total of 1,694 reports (32.0%) contained complete and analyzable time-to-onset data, which served as the basis for this analysis. As shown in Fig 8, the majority of AEs occurred within the first month of treatment: cholestyramine (65 cases, 58.56%), colestipol (29 cases, 58.00%), and colesevelam (71 cases, 41.52%). Notably, despite an initial decline in AE reports after the first treatment month, all three agents exhibited a rebound increase in AE incidence beyond 6 months of continued use. Persistent AE reporting was observed even after 360 days, accounting for 15–16% of total cases across the BASs. Despite the inherent limitations of spontaneous reporting, including potential reporting delays, recall inaccuracies, and the preferential reporting of acute events, the observed

| SOC | PT(Preffered Term) | a | ROR(95% CI) |
|---|---|---|---|
| Gastrointestinal Disorders | Dysphagia | 57 | 21.51(16.52~28.01) |
| Gastrointestinal Disorders | Diarrhoea | 47 | 2.58(1.93~3.45) |
| Respiratory, Thoracic And Mediastinal Disorders | Choking | 37 | 67.44(48.68~93.44) |
| Gastrointestinal Disorders | Constipation | 35 | 5.83(4.17~8.15) |
| Gastrointestinal Disorders | Nausea | 25 | 1.09(0.74~1.62) |
| Gastrointestinal Disorders | Flatulence | 20 | 12.43(8~19.32) |
| General Disorders And Administration Site Conditions | Malaise | 20 | 1.54(0.99~2.39) |
| Gastrointestinal Disorders | Vomiting | 19 | 1.41(0.9~2.22) |
| Gastrointestinal Disorders | Abdominal Distension | 17 | 5.74(3.56~9.25) |
| Gastrointestinal Disorders | Abdominal Discomfort | 17 | 3.55(2.2~5.72) |
| Gastrointestinal Disorders | Abdominal Pain Upper | 16 | 2.72(1.66~4.45) |
| Nervous System Disorders | Dizziness | 14 | 0.96(0.57~1.63) |
| Respiratory, Thoracic And Mediastinal Disorders | Dyspnoea | 14 | 0.85(0.5~1.43) |
| Skin And Subcutaneous Tissue Disorders | Rash | 13 | 1.05(0.61~1.82) |
| Musculoskeletal And Connective Tissue Disorders | Myalgia | 12 | 2.42(1.37~4.27) |
| Immune System Disorders | Drug Hypersensitivity | 12 | 2.08(1.18~3.66) |
| Gastrointestinal Disorders | Dyspepsia | 11 | 3.92(2.17~7.1) |
| Gastrointestinal Disorders | Abdominal Pain | 11 | 1.63(0.9~2.95) |
| General Disorders And Administration Site Conditions | Feeling Abnormal | 11 | 1.53(0.85~2.77) |
| Skin And Subcutaneous Tissue Disorders | Pruritus | 11 | 1.06(0.59~1.92) |
| General Disorders And Administration Site Conditions | Fatigue | 11 | 0.48(0.27~0.88) |
| Respiratory, Thoracic And Mediastinal Disorders | Throat Irritation | 9 | 7.07(3.67~13.6) |
| Investigations | Weight Decreased | 9 | 1.1(0.57~2.12) |
| General Disorders And Administration Site Conditions | Asthenia | 9 | 0.82(0.42~1.57) |
| General Disorders And Administration Site Conditions | Pain | 9 | 0.49(0.25~0.94) |
| Psychiatric Disorders | Insomnia | 8 | 1.01(0.51~2.03) |
| Respiratory, Thoracic And Mediastinal Disorders | Cough | 7 | 0.87(0.41~1.83) |
| Musculoskeletal And Connective Tissue Disorders | Pain In Extremity | 7 | 0.79(0.38~1.67) |
| Nervous System Disorders | Headache | 7 | 0.38(0.18~0.8) |
| Gastrointestinal Disorders | Retching | 6 | 9.78(4.39~21.8) |

**Fig 5. Adverse event signal distribution of colestipol at Preferred Term level.** Top 30 PTs ranked by report frequency. SOC, System Organ Class; a, number of reports containing both the target drug and the target adverse drug reaction; ROR, Reporting Odds Ratio.

biphasic onset pattern across all three BASs suggests a meaningful temporal signal that merits further clinical investigation.

## Discussion

This pharmacovigilance study leveraging real-world data from FAERS elucidated the adverse reaction profiles and potential risk signals of three BASs. Through analysis of deduplicated AE reports, we not only validated the known gastrointestinal adverse effects associated with BAS therapy but also identified novel safety signals not currently documented in drug labels, including respiratory irritation, musculoskeletal symptoms, and mechanical obstruction risks. Importantly, our

| SOC | PT(Preffered Term) | a | ROR(95% CI) |
|---|---|---|---|
| Gastrointestinal Disorders | Constipation | 252 | 10.05(8.86~11.4) |
| Gastrointestinal Disorders | Diarrhoea | 148 | 1.9(1.62~2.24) |
| Gastrointestinal Disorders | Dysphagia | 121 | 10.59(8.85~12.68) |
| Gastrointestinal Disorders | Nausea | 114 | 1.18(0.98~1.42) |
| Musculoskeletal And Connective Tissue Disorders | Myalgia | 99 | 4.74(3.89~5.78) |
| Gastrointestinal Disorders | Abdominal Discomfort | 87 | 4.29(3.47~5.3) |
| Gastrointestinal Disorders | Dyspepsia | 78 | 6.59(5.27~8.24) |
| Musculoskeletal And Connective Tissue Disorders | Muscle Spasms | 78 | 3.43(2.74~4.29) |
| Gastrointestinal Disorders | Abdominal Distension | 65 | 5.17(4.05~6.6) |
| Gastrointestinal Disorders | Abdominal Pain Upper | 65 | 2.6(2.04~3.32) |
| Gastrointestinal Disorders | Vomiting | 61 | 1.06(0.83~1.37) |
| Nervous System Disorders | Headache | 61 | 0.78(0.61~1.01) |
| Gastrointestinal Disorders | Flatulence | 60 | 8.77(6.8~11.31) |
| Nervous System Disorders | Dizziness | 57 | 0.92(0.71~1.2) |
| General Disorders And Administration Site Conditions | Fatigue | 57 | 0.59(0.46~0.77) |
| Musculoskeletal And Connective Tissue Disorders | Pain In Extremity | 50 | 1.34(1.01~1.77) |
| Investigations | Weight Decreased | 49 | 1.42(1.07~1.88) |
| Musculoskeletal And Connective Tissue Disorders | Arthralgia | 48 | 0.95(0.71~1.26) |
| General Disorders And Administration Site Conditions | Malaise | 47 | 0.85(0.64~1.13) |
| General Disorders And Administration Site Conditions | Pain | 47 | 0.6(0.45~0.81) |
| General Disorders And Administration Site Conditions | Asthenia | 46 | 0.98(0.74~1.32) |
| Musculoskeletal And Connective Tissue Disorders | Back Pain | 41 | 1.42(1.04~1.92) |
| General Disorders And Administration Site Conditions | Feeling Abnormal | 41 | 1.34(0.99~1.83) |
| Investigations | Blood Triglycerides Increased | 40 | 18.11(13.26~24.71) |
| Gastrointestinal Disorders | Gastrooesophageal Reflux Disease | 39 | 4.01(2.92~5.49) |
| Immune System Disorders | Drug Hypersensitivity | 35 | 1.43(1.02~1.99) |
| Gastrointestinal Disorders | Abdominal Pain | 32 | 1.12(0.79~1.58) |
| Skin And Subcutaneous Tissue Disorders | Pruritus | 32 | 0.73(0.51~1.03) |
| Investigations | Blood Glucose Increased | 31 | 1.3(0.91~1.84) |
| Respiratory, Thoracic And Mediastinal Disorders | Cough | 30 | 0.88(0.61~1.26) |

**Fig 6. Adverse event signal distribution of colesevelam at Preferred Term level.** Top 30 PTs ranked by report frequency. SOC, System Organ Class; a, number of reports containing both the target drug and the target adverse drug reaction; ROR, Reporting Odds Ratio.

investigation revealed temporal dynamics in long-term medication risks, providing new evidence-based insights for clinical risk management strategies. These findings highlight the critical need for continuous pharmacovigilance surveillance to detect both acute and delayed adverse effects, particularly for medications like BASs that exhibit complex interactions with enterohepatic and metabolic pathways.

Demographic analysis demonstrated that BAS-related AE reports primarily involved female patients and elderly patients aged 65 and above. Existing evidence suggests that estrogen plays a regulatory role in bile acid synthesis and enterohepatic circulation by modulating hepatic cholesterol 7α-hydroxylase (CYP7A1) activity and bile acid transporter expression [24]. The irreversible binding of bile acids by BAS may disproportionately disrupt bile acid homeostasis in females, potentially leading to intestinal microenvironment dysregulation and subsequent gastrointestinal manifestations such as diarrhea and abdominal distension. This gender-specific vulnerability may also be amplified by hormonal fluctuations during

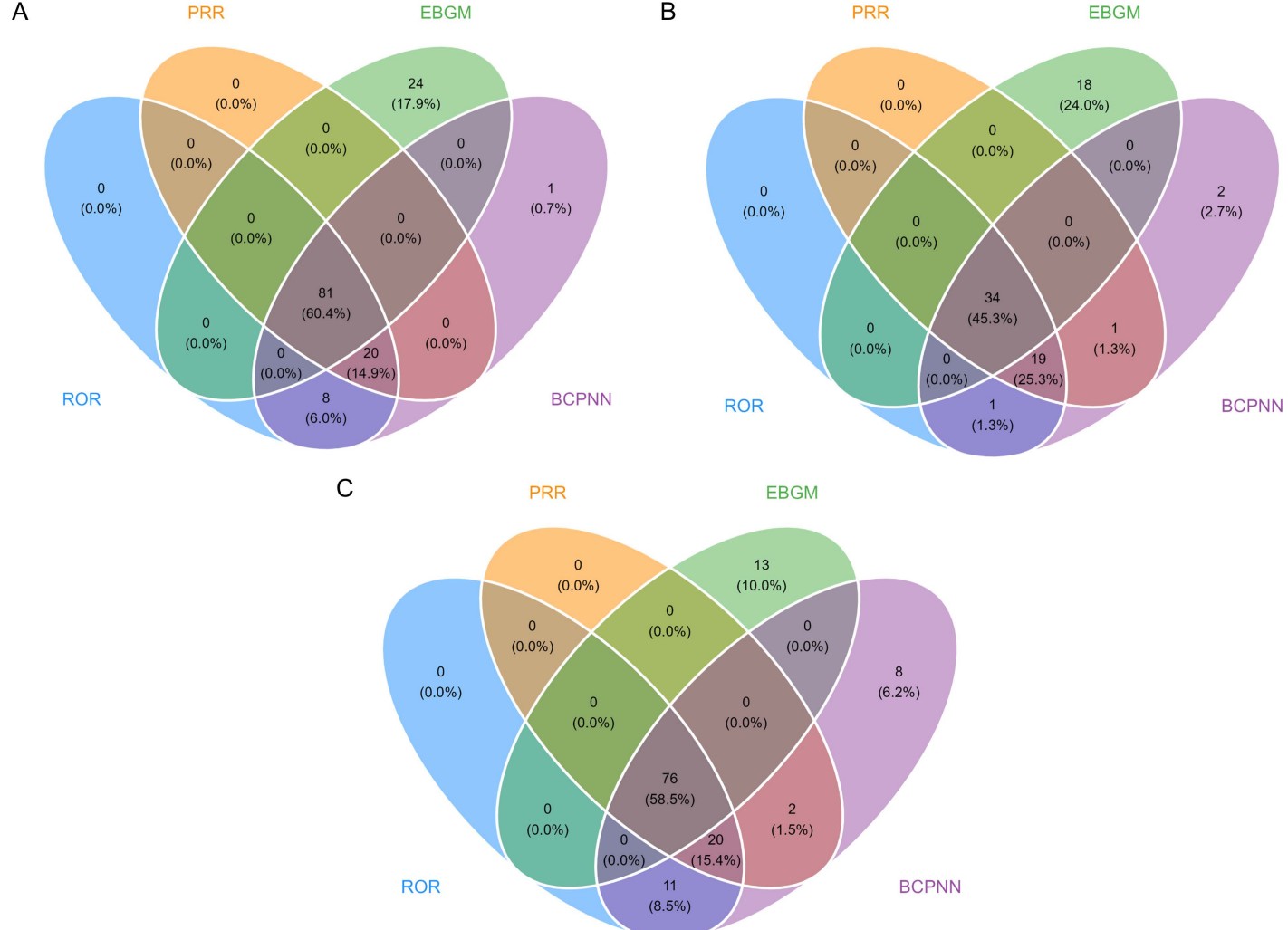

**Fig 7. Consensus adverse event signals of bile acid sequestrants identified by intersection analysis of four disproportionality methods. A: cholestyramine; B: colestipol; C: colesevelam.** ROR, Reporting Odds Ratio; PRR, Proportional Reporting Ratio; BCPNN, Bayesian Confidence Propagation Neural Network; MGPS, Multi-item Gamma Poisson Shrinker.

menstrual cycles or menopause, which could further destabilize bile acid metabolism [25]. Furthermore, the observed female predominance aligns with the epidemiological characteristics of BAS-indicated conditions, including bile acid diarrhea and PBC, both of which have a marked female preponderance in clinical populations [26,27]. The elevated reporting rate among elderly patients corresponds to the frequent clinical application of BAS in age-prevalent diseases such as hypercholesterolemia and type 2 diabetes mellitus [28]. This demographic is particularly susceptible to BAS-related AEs due to age-related physiological changes, including diminished gastrointestinal motility, reduced hepatic metabolic capacity, and compromised intestinal barrier integrity [29]. Additionally, age-associated swallowing difficulties and cognitive impairments may increase the risk of improper drug administration (e.g., inadequate hydration with colestipol tablets), predisposing to mechanical complications such as esophageal obstruction [30]. Polypharmacy, prevalent in the elderly, further compounds these risks through drug-drug interactions (e.g., reduced absorption of levothyroxine or warfarin) and cumulative gastrointestinal burden [31,32]. Although data limitations (e.g., missing information, regional reporting biases)

## A  Cholestyramine

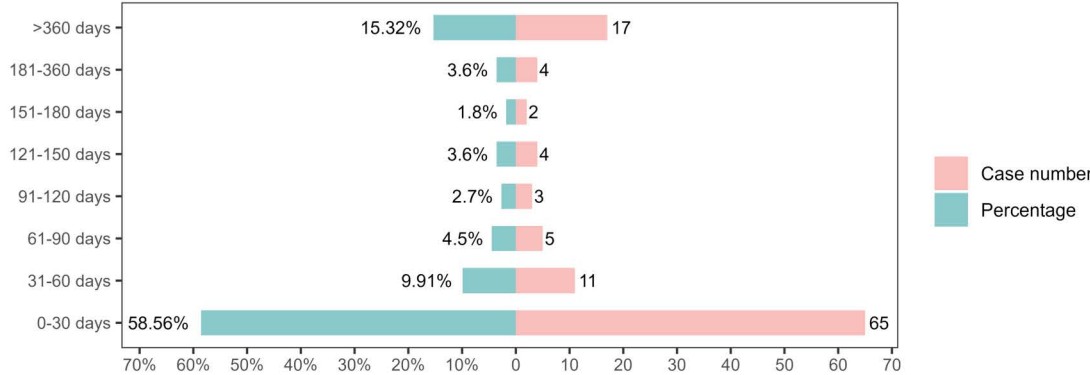

## B  Colestipol

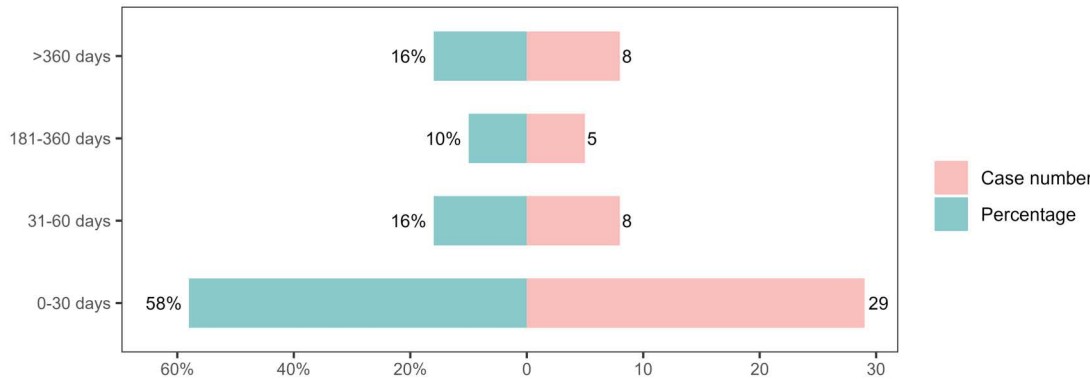

## C  Colesevelam

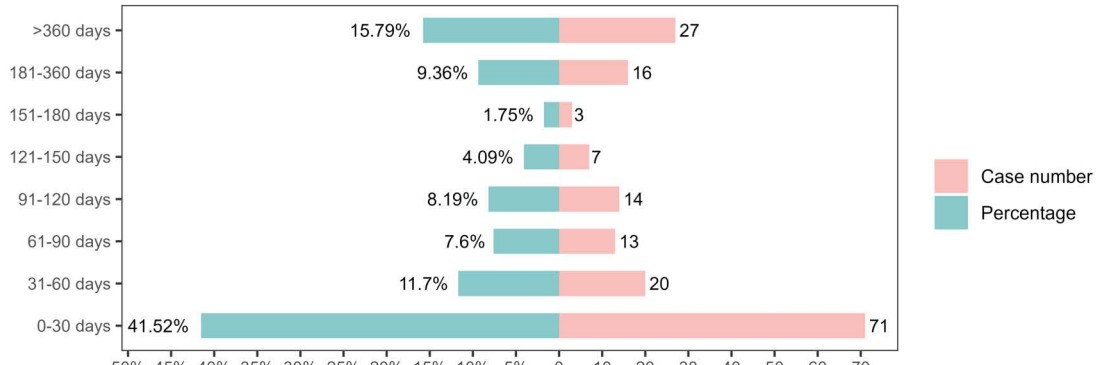

**Fig 8. Time-to-onset analysis of adverse events associated with three bile acid sequestrants.**

might partially exaggerate the observed demographic disparities, current findings underscore the necessity for gender and age-specific monitoring protocols. Prospective studies incorporating pharmacokinetic modeling and frailty indices could refine dosing strategies for elderly patients, while clinical trials stratifying outcomes by hormonal status may optimize BAS use in female populations.

Our analysis revealed consistently high reporting rates of gastrointestinal disorders across all three BAS agents, establishing these as core adverse reactions mainly manifesting as diarrhea, constipation, flatulence, and abdominal pain,

consistent with drug labeling information and existing clinical trial evidence [33,34]. These gastrointestinal events are mechanistically linked to BAS pharmacological actions. By binding bile acids in the intestinal lumen, BAS disrupts entero-hepatic circulation and stimulates compensatory bile acid synthesis, thereby reducing serum cholesterol levels [35]. However, this disruption depletes the bile acid pool necessary for lipid emulsification, leading to steatorrhea and subsequent osmotic diarrhea. Reduced luminal bile acid concentrations may attenuate their physiological stimulation of colonic motility via Takeda G protein-coupled receptor 5 (TGR5), potentially contributing to constipation [36,37]. The gut microbiota plays a pivotal role in BAS-induced dysbiosis [38]. Diminished secondary bile acids (e.g., deoxycholic acid) due to BAS sequestration alter the microbial composition, reducing beneficial taxa while enriching opportunistic pathogens potentially linked to hydrogen sulfide production and abdominal distension [39]. This dysbiotic shift may also impair gut barrier function, increasing intestinal permeability and systemic inflammation—a potential contributor to extraintestinal AEs. Clinicians should routinely inquire about gastrointestinal tolerance in follow-up visits, particularly during the initial weeks of therapy. For patients experiencing persistent diarrhea or steatorrhea, stool fat quantification and nutritional assessment may be warranted. If symptoms suggest BAS-induced dysbiosis, co-administration of probiotics or a gradual dose titration strategy may improve tolerability and adherence.

Notably, this study represents the first FAERS-based identification of significant safety signals in respiratory, neurological, and dermatological SOCs, extending beyond the traditional gastrointestinal-centric safety paradigm for BASs. Drill-down analysis to PT levels showed distinct safety profiles among the three BASs. For instance, cholestyramine demonstrated strong signals for localized irritative symptoms (e.g., pharyngeal irritation, oropharyngeal discomfort), likely attributable to its coarse particulate formulation causing mechanical mucosal abrasion during incomplete dissolution. Case narratives frequently described oral tolerability issues such as "gritty texture" and "throat scraping sensation", suggesting formulation optimization could mitigate these effects [40]. Reports of vitamin K deficiency associated with cholestyramine may relate to its non-selective binding of fat-soluble vitamins, particularly in patients with preexisting malnutrition or prolonged use [15]. This risk underscores the importance of monitoring prothrombin time and supplementing phytonadione in high-risk populations [41]. Periodic coagulation screening (e.g., prothrombin time) should be considered in elderly or malnourished patients on long-term cholestyramine therapy. Where available, fat-soluble vitamin panels (A, D, E, K) could provide a more comprehensive assessment. Prophylactic supplementation of vitamin K may be justified in at-risk individuals, especially those with cholestatic disorders, bariatric history, or concurrent fat malabsorption syndromes. Moderate signals such as dysgeusia and dental abnormalities could stem from micronutrient deficiencies or local oral effects, potentially mitigated by post-dose oral rinsing or utilization of colonic-release formulations [40]. Colestipol exhibited elevated risks of dysphagia and choking, plausibly linked to its large tablet size and hygroscopic expansion properties, predisposing to esophageal retention and mechanical obstruction. Elderly patients with presbyesophagus or structural abnormalities such as Zenker's diverticulum are particularly vulnerable, necessitating pre-treatment swallowing assessments and patient education on proper administration techniques (e.g., adequate water intake, upright positioning). These findings suggest the critical role of formulation optimization and medication guidance for patients in risk mitigation. Colesevelam demonstrated musculoskeletal adverse reaction signals (e.g., myalgia, muscle spasms), which might be indirectly related to electrolyte imbalances (hypomagnesemia/hypocalcemia) and vitamin D malabsorption caused by bile acid homeostasis disruption [42]. Hypomagnesemia-induced neuromuscular hyperexcitability could manifest as muscle cramps, while vitamin D deficiency may exacerbate subclinical myopathy, particularly in patients with baseline nutritional deficits [43]. Baseline screening of serum magnesium, calcium, and 25(OH)D levels may help identify patients at risk for neuromuscular adverse events prior to colesevelam initiation. In cases of unexplained muscle symptoms, early recognition of hypomagnesemia or subclinical myopathy is critical. Electrolyte repletion and vitamin D restoration should be promptly initiated when deficiencies are identified. Novel colesevelam-associated signals validated by multi-algorithm concordance analysis, including fecaloma and gastroesophageal reflux disease, may reflect its polymeric structure's influence on intestinal smooth muscle motility and gastric emptying dynamics [44].

While these mechanistic hypotheses provide potential explanations for the detected AE signals, it is equally important to interpret these findings in a practical context. The clinical relevance and monitoring strategies of such AEs are discussed below. Several novel AE signals identified in this study warrant closer clinical attention due to their potential impact on patient adherence and safety. For instance, throat irritation and oropharyngeal discomfort observed with cholestyramine may result from direct mucosal exposure to undissolved resin granules [45]. These local irritant effects are particularly relevant for elderly patients or those with swallowing difficulties and emphasize the need for proper administration techniques, including adequate dilution and post-dose rinsing. The signal for fecaloma formation associated with colesevelam highlights a rare but serious gastrointestinal complication. Colesevelam's bile acid binding capacity may promote stool dehydration and delayed transit, especially in patients with low fluid intake, reduced mobility, or concurrent use of constipating agents such as calcium or opioid analgesics [46]. Prophylactic measures—such as increasing dietary fiber, fluid intake, and regular bowel monitoring—should be considered for at-risk populations. Another noteworthy signal is dental abnormalities linked to long-term cholestyramine use. While rare, this may reflect prolonged oral contact with the resin, potentially leading to enamel damage or demineralization. This finding suggests a need for dental hygiene counseling, consideration of administration methods (e.g., use of straws), and closer collaboration between prescribing physicians and dental care providers. Collectively, these findings reinforce the importance of tailored monitoring strategies in BAS therapy. Physicians should be aware of both common and underreported AEs, assess patient-specific risk factors, and provide clear guidance on administration and lifestyle modifications to mitigate avoidable harm.

Time-to-onset analysis illustrated a biphasic risk pattern: approximately half of AEs clustered within the first treatment month, likely dependent on BAS pharmacokinetics and the initial intestinal adaptation process. A secondary risk peak emerging after six months of continuous therapy may reflect the cumulative effects of prolonged exposure, such as delayed manifestations of fat-soluble vitamin deficiencies, electrolyte imbalances, or gut microbiota dysbiosis. This bimodal distribution emphasizes the need for differentiated monitoring strategies: prioritizing the prevention of local irritation and acute gastrointestinal reactions during the initiation phase, while paying attention to dynamic surveillance of nutritional and metabolic indices in long-term management.

In light of the observed subclass-specific AE signals, individual variability in bile acid metabolism may underlie differential susceptibility. Genetic polymorphisms in key regulatory genes such as CYP7A1, which encodes cholesterol 7α-hydroxylase, and SLC10A2, responsible for ileal bile acid transport, may modulate drug response and toxicity. Moreover, fibroblast growth factor 19 (FGF19)—a key endocrine regulator of bile acid homeostasis—has been implicated in gastrointestinal and hepatic dysfunction when dysregulated. Although the present study could not assess these biological parameters due to the absence of genotypic and biochemical data in the FAERS database, future pharmacogenomic or translational studies are warranted to validate these mechanistic hypotheses and identify high-risk patient subpopulations.

Several limitations of this study should be noted. First, as a spontaneous reporting system, FAERS is subject to underreporting of mild events and overreporting of rare or severe AEs, potentially introducing reporting bias. Second, the high proportion of missing age data limits the reliability of age-stratified analyses. Third, most reports originated from the United States, restricting generalizability to other populations. Geographic variability in bile acid transporter genetics (e.g., SLC10A2 polymorphisms) and dietary patterns (e.g., fiber intake) may influence both drug efficacy and AE susceptibility [47,48]. Notably, the sharp increase in colesevelam-related reports during 2015–2016 may reflect changes in reporting behavior or periodic follow-up schedules rather than actual increases in drug exposure. Lastly, despite using multiple disproportionality methods to enhance signal robustness, residual confounding from concomitant medications and comorbidities remains unavoidable. Future research incorporating prescription data, electronic health records, and laboratory values will be critical to validate these findings and develop predictive models for BAS-related AEs.

## Conclusions

This real-world pharmacovigilance study identifies distinct safety profiles among BASs through advanced disproportionality analysis. Cholestyramine is linked to localized irritant effects, colestipol demonstrates mechanical obstruction risks, while colesevelam carries a higher propensity for musculoskeletal complaints. Critically, gastrointestinal reactions still constituted the most frequently reported AE class across all three BASs. Novel signals, including dental abnormalities, gastroesophageal reflux disease, and fecaloma, highlight unmet safety concerns. Elderly and female patients exhibited heightened susceptibility, and the time-to-onset analysis showed persistent risks observed beyond 6 months of therapy. These findings necessitate updates to clinical monitoring protocols, emphasizing early intervention for high-risk populations and long-term surveillance. Future research should validate these signals and explore mechanistic pathways to optimize risk-benefit balancing in BAS therapy.

## Supporting information

**S1 Table. Example of the fourfold table for disproportionality analysis in pharmacovigilance.** (XLSX)

**S2 Table. Formulas and signal detection thresholds for ROR, PRR, BCPNN, and MGPS.** Notes: a, number of reports containing both the target drug and the target adverse drug reaction; b, number of reports containing the target drug but associated with other adverse drug reactions; c, number of reports containing the target adverse drug reaction but associated with other drugs; d, number of reports containing other drugs and other adverse drug reactions. Abbreviations: 95%CI, 95% confidence interval; N, the number of reports; $\chi2$, chi-squared; IC, information component; IC025, the lower limit of 95% CI of the IC; E(IC), the IC expectations; V(IC), the variance of IC; EBGM, empirical Bayesian geometric mean; EBGM05, the lower limit of 95% CI of EBGM. (XLSX)

**S3 Table. Gender-stratified adverse reaction signals for bile acid sequestrants.** (XLSX)

**S4 Table. Key adverse reaction signals for bile acid sequestrants in FAERS.** (XLSX)

**S5 Table. Drug-related elevated transaminases in FAERS.** (XLSX)

## Author contributions

**Conceptualization:** Wangqi Chen.

**Data curation:** Wangqi Chen, Qinghua Li.

**Formal analysis:** Wangqi Chen, Yuxia Xie.

**Supervision:** Hong Zhu.

**Writing – original draft:** Wangqi Chen, Yuxia Xie.

**Writing – review & editing:** Wangqi Chen, Yuxia Xie, Qinghua Li, Zhenghui Zhu, Xinyan Li.

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
