## [Decision Letter · Decision Letter 0]

Dear Dr. Zhu,

Thank you for submitting your manuscript to PLOS ONE. After careful consideration, we feel that it has merit but does not fully meet PLOS ONE’s publication criteria as it currently stands. Therefore, we invite you to submit a revised version of the manuscript that addresses the points raised during the review process.

We look forward to receiving your revised manuscript.

Kind regards,

Sharon DeMorrow

Academic Editor

PLOS ONE

Journal Requirements:

Additional Editor Comments:

The reviewers all found merit and novelty in the study, although some significant deficiencies have been noted. Please carefully consider and respond to all of the concerns raised.

Reviewers' comments:

Reviewer's Responses to Questions

**Comments to the Author**

1. Is the manuscript technically sound, and do the data support the conclusions?

Reviewer #1: Yes

Reviewer #2: Yes

Reviewer #3: Partly

Reviewer #4: Yes

Reviewer #5: Yes

2. Has the statistical analysis been performed appropriately and rigorously?

Reviewer #1: Yes

Reviewer #2: Yes

Reviewer #3: I Don't Know

Reviewer #4: Yes

Reviewer #5: Yes

3. Have the authors made all data underlying the findings in their manuscript fully available?

Reviewer #1: Yes

Reviewer #2: Yes

Reviewer #3: Yes

Reviewer #4: Yes

Reviewer #5: Yes

4. Is the manuscript presented in an intelligible fashion and written in standard English?

Reviewer #1: Yes

Reviewer #2: Yes

Reviewer #3: Yes

Reviewer #4: Yes

Reviewer #5: Yes

Reviewer #1: The reviewers acknowledged the novelty and relevance of the study but pointed out several key issues that require revision. They emphasized the need for a clearer structure in the abstract and suggested that methodological details, such as drug selection and signal thresholds, should be elaborated. Reviewers also requested a more precise explanation of the rationale behind choosing the four disproportionality analysis methods. Additionally, they highlighted that while some interesting findings were reported, the clinical interpretation of these adverse events was lacking and should be better contextualized. They encouraged the authors to strengthen the discussion, especially regarding implications for clinical practice, and to ensure consistency in terminology and formatting throughout the manuscript.

Reviewer #2: Bile acid sequestrants are relatively old drugs, originally developed to lower plasma cholesterol to treat hypercholesterolemia-associated cardiovascular disease. The bile acid sequestrants are non-absorbable bindings resins and are widely considered safe, with few associated serious adverse events. In addition to bile acids, bile acid sequestrants will bind a variety of other small molecules including more than 50 other medications, necessitating administration of the other medications well before or after taking cholestyramine. The gastrointestinal adverse events are widely recognized. In addition, patient discomfort with consuming the large quantities of cholestyramine required for therapeutic benefit is likely responsible for reduced patient compliance and efficacy versus the lipid-lowering drugs (such as statins) developed later. Bile acid sequestrants have long been known increase plasma triglyceride levels and contra-indicated in patients with hypertriglyceridemia. Although not associated with clinically apparent acute liver injury, use of bile acid sequestrants is associated with a low rate of mild (1 to 3-fold) serum aminotransferase and alkaline phosphatase elevations which are self-limited and not associated with symptoms or jaundice.

Publication of data regarding adverse events associated with bile acid sequestrant use have mainly been reports of clinical trials for these agents. There is less published information regarding the real-world data for adverse events. In the present study, the authors interrogate the FDA Adverse Event Reporting System (2004 - 2024) to capture additional insights to incidence and types of adverse events.

Specific Comments for the authors

1. There was a dramatic increase in reports for Colesevelam in 2015 and 2016. What accounts for that more than 10-fold increase? Were the number of prescriptions significantly increased? Was there a change in how the FAERS collected data on AEs? Colesevelam was approved by the FDA for the treatment of hypercholesterolemia in 2000 and for hyperglycemia in 2008, well before this spike in reported AEs.

2. The absence of reports of elevated transaminases was surprising. Is there a system that collects data on potential drug-induced liver injury (DILI) that is separate from the FAERS?

Reviewer #3: In this manuscript, the authors presented the data from the FDA Adverse Event Reporting System for the 3 bile acid sequestrants. The data confirmed what have been reported for the major adverse effects of bile acid sequestrants, gastrointestinal symptoms. Although some new minor adverse events (AE) were reported, there is a lack of detailed information on the severity, patient medical conditions, and etc. Therefore, lack of novelty of the study is a major concern. There are many missing pieces of information from the dataset, poor quality of the data sources is another concern.

There are some other specific concerns:

1) The font of Fig. 4 and Fig. 5 is too small to read, which makes it difficult to evaluate the data;

2) In Fig. 2, what cause a sudden increase in AE in 2015 for colesevelam?

3) In Fig. 6 for the time-to-onset analysis, what are short-term AEs? and what are the long-term AEs?

4) The authors should provide information on the rates of AEs, such as 1 in 100 or 10,000.

Reviewer #4: Overall, the manuscript is well-organized, methodologically sound, and contributes valuable real-world evidence to the drug safety literature. However, key aspects—including the clinical interpretation of novel signals, potential confounding factors, and the handling of incomplete demographic and temporal data—require clarification.

1. Clarity of Objectives and Hypotheses: The introduction would benefit from a clearer statement of the study's primary objectives or hypotheses. It is somewhat unclear whether the goal is to update known AE profiles, identify novel signals, or compare subclass-specific risks.

2. Attribution of Causality: Given the inherent limitations of spontaneous reporting systems, how do you differentiate between AEs likely caused by the drug versus those due to underlying diseases (e.g., diabetes, PBC)? Please discuss this challenge in more depth.

3. Novel Signal Interpretation: Several AE signals reported as "novel" (e.g., throat irritation, fecaloma, dental abnormalities) are intriguing. However, their clinical relevance and causality are unclear. Could you elaborate on possible mechanisms or supporting literature?

4. Time-to-Onset Analysis: The biphasic AE onset pattern is a compelling finding. However, FAERS often lacks detailed dosing and duration data. Please clarify how many reports included complete and reliable onset data, and discuss potential biases in this analysis.

5. Handling of Demographic Gaps: A substantial proportion of reports lack age or sex data. While this limitation is acknowledged, it may affect subgroup analyses. Have you considered performing sensitivity analyses to evaluate whether your findings are robust despite these missing data?

6. Clinical Impact and Recommendations: Based on your findings, what specific changes, if any, would you recommend to clinical practice or patient monitoring? Clarifying the real-world implications of your work would strengthen the conclusion.

Reviewer #5: I have the following comments for the manuscript's authors: "Update of safety profile of bile acid sequestrants: A real-world pharmacovigilance study of the FDA adverse event reporting system."

• The author should provide information on the SNPs in the CYP7A1 gene among the Cholestyramine, Colestipol, and Colesevelam treatment groups showing symptoms like dental abnormalities, gastroesophageal reflux disease, and fecaloma.

• The authors should provide the association of plasma FGF19 levels among the Cholestyramine, Colestipol, and Colesevelam treatment groups showing symptoms like dental abnormalities, gastroesophageal reflux disease, and fecaloma.

**Do you want your identity to be public for this peer review?** For information about this choice, including consent withdrawal, please see our Privacy Policy

Reviewer #1: No

Reviewer #2: No

Reviewer #3: No

Reviewer #4: No

Reviewer #5: No

---

## [Author Response · Author response to Decision Letter 1]

20 Jun 2025

Dear reviewers:

Thank you very much for your comments and professional advice. These opinions help to improve the academic rigor of our article. Based on your suggestion and request, we have made the correct modifications to the revised manuscript. We hope that our work can be improved again. Furthermore, we would like to show the details as follows:

Reviewer #1:

The reviewers acknowledged the novelty and relevance of the study but pointed out several key issues that require revision. They emphasized the need for a clearer structure in the abstract and suggested that methodological details, such as drug selection and signal thresholds, should be elaborated.

Response: We appreciate the reviewer’s suggestion. While the abstract follows a standard structured format (Background, Methods, Results, Conclusion), we acknowledge that the high information density may have affected readability. To address this, we revised the abstract for greater conciseness and improved clarity. Redundant phrasing was reduced, and key results were streamlined for easier interpretation. As for methodological details, we agree and have significantly expanded the “Materials and methods” section to clarify methodological choices (line164-170). We now provide a detailed explanation of the drug selection criteria, including reliance on FDA-approved BASs, incorporation of both generic and brand names, and restriction to reports listing BASs as Primary Suspect Drugs (PS) to ensure causality relevance. We also added the precise thresholds for signal detection across the four disproportionality methods (ROR, PRR, BCPNN, MGPS), and summarized these definitions in both the main text and Supplementary Table 2.

Reviewers also requested a more precise explanation of the rationale behind choosing the four disproportionality analysis methods.

Response: Thank you for this suggestion. We now explicitly state the rationale behind choosing the four algorithms (line189-199). ROR and PRR were chosen as established frequentist approaches suitable for high-throughput signal generation. BCPNN and MGPS were selected to account for Bayesian prior correction, particularly valuable in sparse or low-frequency data scenarios. The combined use enhances signal validity and reduces the risk of spurious findings.

Additionally, they highlighted that while some interesting findings were reported, the clinical interpretation of these adverse events was lacking and should be better contextualized. They encouraged the authors to strengthen the discussion, especially regarding implications for clinical practice, and to ensure consistency in terminology and formatting throughout the manuscript.

Response: We appreciate the reviewer’s insightful suggestion to improve the clinical contextualization of our findings. In the revised version, we have significantly expanded the discussion section (line385-390, line404-409, line426-430, and line436-459) to provide practical interpretations of the observed adverse events (AEs), particularly regarding monitoring strategies and management recommendations.

Specifically, we discuss the real-world relevance of novel signals such as throat irritation, fecaloma formation, and dental abnormalities. These discussions now include possible mechanisms, high-risk populations (e.g., elderly individuals with dysphagia or poor dentition), and actionable strategies such as proper administration techniques, dental hygiene education, and bowel function monitoring.

In addition, we have revised the paragraphs on gastrointestinal, musculoskeletal, and vitamin-related adverse events to incorporate detailed clinical suggestions. These include the use of stool fat quantification and vitamin panels in patients with persistent diarrhea, periodic prothrombin time testing and vitamin K supplementation in malnourished populations, and baseline electrolyte screening for patients treated with colesevelam.

These revisions aim to bridge the gap between signal detection and clinical decision-making, aligning with the reviewer’s recommendation. We believe these additions enhance the translational value of our findings and their relevance to patient safety and pharmacovigilance.

Reviewer #2:

Bile acid sequestrants are relatively old drugs, originally developed to lower plasma cholesterol to treat hypercholesterolemia-associated cardiovascular disease. The bile acid sequestrants are non-absorbable bindings resins and are widely considered safe, with few associated serious adverse events. In addition to bile acids, bile acid sequestrants will bind a variety of other small molecules including more than 50 other medications, necessitating administration of the other medications well before or after taking cholestyramine. The gastrointestinal adverse events are widely recognized. In addition, patient discomfort with consuming the large quantities of cholestyramine required for therapeutic benefit is likely responsible for reduced patient compliance and efficacy versus the lipid-lowering drugs (such as statins) developed later. Bile acid sequestrants have long been known increase plasma triglyceride levels and contra-indicated in patients with hypertriglyceridemia. Although not associated with clinically apparent acute liver injury, use of bile acid sequestrants is associated with a low rate of mild (1 to 3-fold) serum aminotransferase and alkaline phosphatase elevations which are self-limited and not associated with symptoms or jaundice.

Publication of data regarding adverse events associated with bile acid sequestrant use have mainly been reports of clinical trials for these agents. There is less published information regarding the real-world data for adverse events. In the present study, the authors interrogate the FDA Adverse Event Reporting System (2004 - 2024) to capture additional insights to incidence and types of adverse events.

Specific Comments for the authors

1.There was a dramatic increase in reports for Colesevelam in 2015 and 2016. What accounts for that more than 10-fold increase? Were the number of prescriptions significantly increased? Was there a change in how the FAERS collected data on AEs? Colesevelam was approved by the FDA for the treatment of hypercholesterolemia in 2000 and for hyperglycemia in 2008, well before this spike in reported AEs.

Response: We sincerely thank the reviewer for raising this important observation regarding the dramatic increase in colesevelam-related adverse event (AE) reports during 2015 and 2016. We acknowledge that this temporal spike, as shown in Figure 2 of the manuscript, does not correspond with the drug’s FDA approval timeline (2000 for hypercholesterolemia and 2008 for hyperglycemia), and thus warrants careful exploration.

To investigate this pattern, we reviewed U.S. prescription data from ClinCalc DrugStats, which is based on Medicare Part D claims and reflects outpatient medication use among the elderly population. These data indicate that the number of prescriptions for bile acid sequestrants—including colesevelam—actually declined significantly over this period. Although this dataset does not represent the entire U.S. population, the observed downward trend suggests that the spike in FAERS reports is unlikely to be driven by increased drug exposure.

We therefore believe that the increase in AE reporting may be attributed to changes in pharmacovigilance behavior. For example, heightened safety monitoring efforts by the manufacturer or regulatory bodies, increased clinician awareness, or broader use of colesevelam in combination regimens for complex metabolic disorders may have contributed to elevated reporting during that period. Furthermore, given that colesevelam is typically used in long-term management of chronic conditions such as hyperlipidemia and type 2 diabetes, many AEs—such as constipation, musculoskeletal complaints, or vitamin deficiencies—are more likely to manifest or be recognized during periodic follow-up visits rather than immediately after drug initiation. It is therefore plausible that AE submissions were temporally clustered around routine clinical review cycles (e.g., every 3–6 months), resulting in apparent spikes in the FAERS database.

While FAERS does not contain detailed prescription or visit-level data to directly verify this hypothesis, we appreciate the opportunity to clarify this limitation. We have not modified the main text but acknowledge that future studies integrating real-world drug utilization databases (e.g., IQVIA, MEPS, or Medicare claims) may better contextualize temporal reporting patterns observed in spontaneous reporting systems.

2.The absence of reports of elevated transaminases was surprising. Is there a system that collects data on potential drug-induced liver injury (DILI) that is separate from the FAERS?

Response: We thank the reviewer for this insightful comment. To clarify, we conducted a focused retrieval of adverse event reports related to elevated transaminases, specifically the Preferred Term “alanine aminotransferase increased.” The results are presented in Supplementary Table 12, which lists all drugs with ten or more associated reports.

Importantly, none of the three bile acid sequestrants were among the drugs meeting this threshold, indicating that elevated transaminase events linked to these agents are either infrequent or underreported in the FAERS database. This observation aligns with the nature of spontaneous reporting systems, where laboratory abnormalities without clinical manifestations are less likely to be captured.

Reviewer #3:

In this manuscript, the authors presented the data from the FDA Adverse Event Reporting System for the 3 bile acid sequestrants. The data confirmed what have been reported for the major adverse effects of bile acid sequestrants, gastrointestinal symptoms. Although some new minor adverse events (AE) were reported, there is a lack of detailed information on the severity, patient medical conditions, and etc. Therefore, lack of novelty of the study is a major concern. There are many missing pieces of information from the dataset, poor quality of the data sources is another concern.

There are some other specific concerns:

1)The font of Fig. 4 and Fig. 5 is too small to read, which makes it difficult to evaluate the data;

Response: We thank the reviewer for pointing this out. In response, we have revised Figure 4 by splitting it into separate panels (Figure 4,5,6) to improve clarity and ensure that all labels and legends are legible. Additionally, Figure 5 has been enlarged and reformatted to enhance font size and visual readability. We believe these changes significantly improve data interpretation, and we appreciate the reviewer’s suggestion.

2)In Fig. 2, what cause a sudden increase in AE in 2015 for colesevelam?

Response: We sincerely thank the reviewer for raising this important observation regarding the dramatic increase in colesevelam-related adverse event (AE) reports during 2015 and 2016. We acknowledge that this temporal spike, as shown in Figure 2 of the manuscript, does not correspond with the drug’s FDA approval timeline (2000 for hypercholesterolemia and 2008 for hyperglycemia), and thus warrants careful exploration.

To investigate this pattern, we reviewed U.S. prescription data from ClinCalc DrugStats, which is based on Medicare Part D claims and reflects outpatient medication use among the elderly population. These data indicate that the number of prescriptions for bile acid sequestrants—including colesevelam—actually declined significantly over this period. Although this dataset does not represent the entire U.S. population, the observed downward trend suggests that the spike in FAERS reports is unlikely to be driven by increased drug exposure.

We therefore believe that the increase in AE reporting may be attributed to changes in pharmacovigilance behavior. For example, heightened safety monitoring efforts by the manufacturer or regulatory bodies, increased clinician awareness, or broader use of colesevelam in combination regimens for complex metabolic disorders may have contributed to elevated reporting during that period. Furthermore, given that colesevelam is typically used in long-term management of chronic conditions such as hyperlipidemia and type 2 diabetes, many AEs—such as constipation, musculoskeletal complaints, or vitamin deficiencies—are more likely to manifest or be recognized during periodic follow-up visits rather than immediately after drug initiation. It is therefore plausible that AE submissions were temporally clustered around routine clinical review cycles (e.g., every 3–6 months), resulting in apparent spikes in the FAERS database.

While FAERS does not contain detailed prescription or visit-level data to directly verify this hypothesis, we appreciate the opportunity to clarify this limitation. We have not modified the main text but acknowledge that future studies integrating real-world drug utilization databases (e.g., IQVIA, MEPS, or Medicare claims) may better contextualize temporal reporting patterns observed in spontaneous reporting systems.

3)In Fig. 6 for the time-to-onset analysis, what are short-term AEs? and what are the long-term AEs?

Response: We thank the reviewer for this important question. We used a 30-day threshold to distinguish short-term versus long-term adverse events (AEs) based on its wide adoption in pharmacovigilance research and clinical pharmacology. Previous studies have shown that most acute, dose-related adverse events occur within the first month of treatment, and 30 days is a commonly used risk window in spontaneous reporting systems such as FAERS. This threshold also aligns with signal detection methodologies employed by regulatory agencies and enhances comparability across drug safety studies.

Reference: Jiang P, Zong K, Peng D, Zhou B, Wu Z. Risk comparison of adverse reactions between gemcitabine monotherapy and gemcitabine combined with albumin-bound paclitaxel in pancreatic cancer: insights from the FDA Adverse Event Reporting System (FAERS) database. BMC Pharmacol Toxicol. 2025 Mar 19;26(1):65. doi: 10.1186/s40360-025-00884-5. PMID: 40108669; PMCID: PMC11924625.

4)The authors should provide information on the rates of AEs, such as 1 in 100 or 10,000.

Response: We thank the reviewer for this valuable suggestion. In response, we have now calculated the estimated reporting rates of adverse events based on available drug utilization data. These rates have been added to the Results section and summarized in Supplementary Table 9-11 for clarity. We hope this addition enhances the interpretability and clinical relevance of our findings.

Reviewer #4:

Overall, the manuscript is well-organized, methodologically sound, and contributes valuable real-world evidence to the drug safety literature. However, key aspects—including the clinical interpretation of novel signals, potential confounding factors, and the handling of incomplete demographic and temporal data—require clarification.

1.Clarity of Objectives and Hypotheses: The introduction would benefit from a clearer statement of the study's primary objectives or hypotheses. It is somewhat unclear whether the goal is to update known AE profiles, identify novel signals, or compare subclass-specific risks.

Response: We appreciate the reviewer’s insightful comment. We have revised the Introduction section (line123-133) to clarify the primary aims of this study: (1) to update and quantify known adverse event (AE) profiles of bile acid sequestrants (BASs), (2) to identify potentially novel AE signals, and (3) to compare subclass-specific risk patterns among cholestyramine, colestipol, and colesevelam using real-world pharmacovigilance data.

2.Attribution of Causality: Given the inherent limitations of spontaneous reporting systems, how do you differentiate between AEs likely caused by the drug versus those due to underlying diseases (e.g., diabetes, PBC)? Please discuss this challenge in more depth.

Response: We fully acknowledge this important limitation. As noted in the revised Discussion section, FAERS is a spontaneous reporting system and cannot establish definitive causality. Confounding from underlying diseases (e.g., diabetes, PBC) and concomitant medications may influence AE o

---

## [Decision Letter · Decision Letter 1]

Update of safety profile of bile acid sequestrants: A real-world pharmacovigilance study of the FDA adverse event reporting system

PONE-D-25-16626R1

Dear Dr. Zhu,

We’re pleased to inform you that your manuscript has been judged scientifically suitable for publication and will be formally accepted for publication once it meets all outstanding technical requirements.

Kind regards,

Sharon DeMorrow

Academic Editor

PLOS ONE

Additional Editor Comments (optional):

Reviewers' comments:

Reviewer's Responses to Questions

**Comments to the Author**

Reviewer #2: All comments have been addressed

Reviewer #3: All comments have been addressed

Reviewer #4: All comments have been addressed

Reviewer #5: All comments have been addressed

2. Is the manuscript technically sound, and do the data support the conclusions?

Reviewer #2: Yes

Reviewer #3: Yes

Reviewer #4: Yes

Reviewer #5: Yes

3. Has the statistical analysis been performed appropriately and rigorously?

Reviewer #2: Yes

Reviewer #3: Yes

Reviewer #4: Yes

Reviewer #5: Yes

4. Have the authors made all data underlying the findings in their manuscript fully available?

Reviewer #2: Yes

Reviewer #3: Yes

Reviewer #4: Yes

Reviewer #5: Yes

5. Is the manuscript presented in an intelligible fashion and written in standard English?

Reviewer #2: Yes

Reviewer #3: Yes

Reviewer #4: Yes

Reviewer #5: Yes

Reviewer #2: (No Response)

Reviewer #3: The authors has done great job in addressing my comments. I have no major concerns. However, the font for Fig. 7 is still too small to be read clearly.

Reviewer #4: Thank you for your careful attention to the reviewers’ feedback. All concerns appear to have been addressed appropriately.

Reviewer #5: The authors of the manuscript "Update of safety profile of bile acid sequestrants: A real-world pharmacovigilance study of the FDA adverse event reporting system" have addressed the comments to my satisfaction.

**Do you want your identity to be public for this peer review?** For information about this choice, including consent withdrawal, please see our Privacy Policy

Reviewer #2: No

Reviewer #3: No

Reviewer #4: **Yes: ** Pradeep Kumar

Reviewer #5: No

---

## [Editor Report · Acceptance letter]

PONE-D-25-16626R1

PLOS ONE

Dear Dr. Zhu,

I'm pleased to inform you that your manuscript has been deemed suitable for publication in PLOS ONE. Congratulations! Your manuscript is now being handed over to our production team.

Kind regards,

on behalf of

Dr. Sharon DeMorrow

Academic Editor

PLOS ONE